# Iron trafficking in patients with Indian Post kala-azar dermal leishmaniasis

**Aishwarya Dighal[1‡], Debanjan Mukhopadhyay [1‡], Ritika Sengupta[1], Srija Moulik[1], Shibabrata Mukherjee [1], Susmita Roy[1], Surya Jyati Chaudhuri[2], Nilay K. Das[3], Mitali Chatterjee [1]***

**1** Dept. of Pharmacology, Institute of Postgraduate Medical Education and Research, Kolkata, India, **2** Dept of Microbiology, Govt. Medical College, Purulia, India, **3** Dept of Dermatology, Bankura Sammilani Medical College, Bankura, India

‡ These authors share first authorship on this work.
* ilatimc@gmail.com

**Data Availability Statement:** All relevant data are within the manuscript and its Supporting Information files.

## Abstract

### Background

During infections involving intracellular pathogens, iron performs a double-edged function by providing the pathogen with nutrients, but also boosts the host's antimicrobial arsenal. Although the role of iron has been described in visceral leishmaniasis, information regarding its status in the dermal sequel, Post Kala-azar Dermal Leishmaniasis (PKDL) remains limited. Accordingly, this study aimed to establish the status of iron within monocytes/macrophages of PKDL cases.

### Methodology/Principal findings

The intramonocytic labile iron pool (LIP), status of CD163 (hemoglobin-haptoglobin scavenging receptor) and CD71 (transferrin receptor, Tfr) were evaluated within CD14$^+$ monocytes by flow cytometry, and soluble CD163 by ELISA. At the lesional sites, Fe$^{3+}$ status was evaluated by Prussian blue staining, parasite load by qPCR, while the mRNA expression of Tfr (*TfR1/CD71*), *CD163*, divalent metal transporter-1 (*DMT-1*), Lipocalin-2 (*Lcn-2*), Heme-oxygenase-1 (*HO-1*), Ferritin, Natural resistance-associated macrophage protein (*NRAMP-1*) and Ferroportin (*Fpn-1*) was evaluated by droplet digital PCR. Circulating monocytes demonstrated elevated levels of CD71, CD163 and soluble CD163, which corroborated with an enhanced lesional mRNA expression of *TfR*, *CD163*, *DMT1* and *Lcn-2*. Additionally, the LIP was raised along with an elevated mRNA expression of *ferritin* and *HO-1*, as also iron exporters *NRAMP-1* and *Fpn-1*.

### Conclusions/Significance

In monocytes/macrophages of PKDL cases, enhancement of the iron influx gateways (TfR, CD163, DMT-1 and Lcn-2) possibly accounted for the enhanced LIP. However, enhancement of the iron exporters (NRAMP-1 and Fpn-1) defied the classical Ferritin$^{low}$/Ferroportin$^{high}$ phenotype of alternatively activated macrophages. The creation of such a pro-parasitic

**Funding:** The work received financial assistance from Indian Council of Medical Research (ICMR), Govt. of India [Grant number: 6/9-7(151)2017-ECD II], Department of Health Research (DHR), Govt. of India [Grant number: DHR/HRD/Fellowship/SUG-05/2015-16], Fund for Improvement of S&T infrastructure in Universities and Higher Educational Institutions (FIST) Program, Dept. of Science & Technology, Govt. of India (DST-FIST) [Grant number: SR/FST/LSI-663/2016] Dept. of Science & Technology, Govt. of West Bengal [Grant number: 969(Sanc.)/ST/P/S&T/9G-22/2016] and Multidisciplinary Research Unit (MRU), Department of Health Research (DHR), Govt. of India [Grant number: V.25011/611/2016-HR]. AD is a recipient of a Junior Research Fellowship from University Grants Commission, DM and RS are recipients of a Senior and Junior Research Fellowship respectively from ICMR, Govt. of India, S Mukherjee is a recipient of a Senior Research Fellowship from INSPIRE Programme DST, Govt. of India and SR is a recipient of a Senior Research Fellowship from CSIR, Govt. of India. The funders had no role in study design, data collection and analysis, decision to publish, or preparation of the manuscript.

**Competing interests:** The authors have declared that no competing interests exist.

environment suggests incorporation of chemotherapeutic strategies wherein the availability of iron to the parasite can be restricted.

## Author summary

Post kala-azar dermal leishmaniasis (PKDL), a dermal sequel of Visceral Leishmaniasis (VL) is caused by the digenetic protozoan parasite *Leishmania donovani*. The parasite infects humans and replicates intracellularly within macrophages, cells normally engaged in protecting the host from pathogens. Iron plays a crucial role in microbes and mammalian cells, being needed by the former for its growth and survival, while the latter uses it for activation of the immune system by facilitating generation of reactive oxygen species. Therefore, the availability of iron needs to be tightly regulated to ensure its accessibility for core biological functions, and yet prevent its utilization by intracellular pathogens. Here we investigated the status of intra-macrophage iron along with expression of its transporters in patients with PKDL. Our study suggests that within monocytes/macrophages there is an enhanced entry of iron via the upregulation of CD71 and CD163 that translates into an enhanced labile iron pool and *Ferritin*. However, the concomitant increase in expression of iron exporters *NRAMP-1* and *Fpn-1* suggested the host's attempt to deny the pathogen access to iron. This Ferritin$^{high}$/Ferroportin$^{high}$ phenotype was in contrast to the conventional Ferritin$^{low}$/Ferroportin$^{high}$ phenotype present in alternatively activated M2 macrophages. Taken together, the control of iron homeostasis is one of the contributors in the host-pathogen interplay as it influences the course of an infectious disease by favouring either the mammalian host or the invading pathogen.

## Introduction

Leishmaniases is caused by the intracellular, digenetic protozoan parasite *Leishmania* that replicates within phagolysosomes of host macrophages. The diverse disease spectrum is attributed to the multiple species that can cause self healing cutaneous lesions, non healing muco-cutaneous lesions involving the mucosa or have visceral involvement of the liver and spleen and cause kala-azar or Visceral Leishmaniasis (VL), which in some apparently cured cases manifests as a dermal sequel, Post Kala-azar Dermal Leishmaniasis (PKDL) [1]. The survival of this intracellular pathogen within host macrophages relies on its ability to effectively nullify host microbicidal effector mechanisms [2], and thrive within acidified, hydrolase-rich phagolysosomes which conventionally constitute compartments responsible for elimination of invading pathogens [3].

An important evolutionary adaptation in parasites is their acquisition of essential nutrients from host cells [4], which includes iron, a trace element essential for virtually all forms of life, as it functions as a cofactor of metabolic enzymes, oxygen transport and participates in immune surveillance [5]. Accordingly, intracellular pathogens deploy several strategies for iron acquisition from host macrophages [6,7] to ensure their intracellular growth [8, 9]. However they also need to minimize the host's oxidative stress response where iron is a cofactor for superoxide dismutase (Fe-SOD) [10], and therefore, its inactivation is essential for their intracellular survival [11].

In view of the absence of an animal model for PKDL, the role of iron, if any, remains poorly defined. Accordingly, this study was undertaken in patients with PKDL, representative of a

chronic manifestation of Leishmanisis, with a view to delineate within circulating monocytes and lesional monocytes-macrophages, the acquisition and export of iron, along with the status of iron metabolism with a view to designing chemotherapeutic strategies that can potentially limit the availability of iron to *Leishmania* parasites.

## Materials and methods

### Chemicals

All antibodies were from BD Biosciences (San Jose, CA, USA) and reagents from Sigma Aldrich (St. Louis, MO, USA), except rK39 immunochromatographic test strips (InBios International, Seattle, WA, USA), QIAmp DNA Mini kit (Qiagen, Hilden, Germany), SYBR Green qPCR Master Mix (Applied Biosystems, Grand Island, NY, USA), cDNA Reverse Transcription kit (Applied Biosystems, Grand Island, NY, USA), anti-human CD68 (clone PG-M1), secondary detection system EnVision G|2 System/AP-Rabbit/Mouse (Permanent Red), EnVision FLEX Target Retrieval Solution (Dako, Glostrup, Denmark), and CD163 kit (RayBiotech, Norcross, GA, USA). All reagents, instruments and analysing software for droplet digital PCR were from Bio-Rad Laboratories (Hercules, CA, USA).

### Study population

Patients clinically diagnosed with PKDL (n = 25) were recruited either from the Dermatology outpatient departments of School of Tropical Medicine/Calcutta Medical College/Institute of PG Medical Education & Research, Kolkata, West Bengal or from active field surveys conducted in endemic districts of West Bengal (Malda, Dakshin Dinajpur, Murshidabad and Birbhum) by a camp approach, wherein a door-to-door survey was conducted by Kala-azar Technical Supervisors using standard case definitions and defined risk factors e.g. living in an endemic area and having an epidemiological link (past history of VL) as also a rK39 strip test positivity [12].

   These suspected cases were examined at medical camps; cases with hypopigmented macules were considered as macular PKDL, whereas an assortment of papules, nodules, macules, and/or plaques were considered as polymorphic [13, 14]; diagnosis was confirmed by ITS-1 PCR from skin biopsies [15]. Skin biopsies from healthy individuals (n = 3) undergoing voluntary circumcision were taken for the ddPCR based work; for other experiments, age and sex-matched healthy volunteers (n = 24) were recruited from endemic and non-endemic areas and were seronegative for anti-leishmanial antibodies as tested by ELISA. None suffered from any co-infection or pre-existing disease.

### Measurement of parasite load by real time PCR

For measurement of parasite load, a standard curve was generated by adding a defined number of *L. donovani* parasites (MHOM/IN/1983/AG83) ranging from 10 to 1 X $10^5$ to blood (180 μl) from a healthy control [15]. Real-time PCR was performed using specific primers for minicircle kDNA (116 bp, forward 5'-CCTATTTTACACCAACCCCCAGT-3'and reverse 5'GGGT AGGGGCGTTCTGCGAAA-3' [16]. DNA (1 μl) was added to a 19 μl reaction mixture containing SYBR Green Master mix and 400 nM of each primer. Negative controls used were DNA from a healthy donor (no amplification), and a reaction mixture with water instead of template DNA (No template control, NTC). The number of parasites was extrapolated from the standard curve and final parasite load stated as the number/μg of genomic DNA. The parasite number when <10 reported a $C_t$ value equivalent to NTC and was accorded an arbitrary value of 1 [16].

## Measurement of plasma iron and sCD163

Plasma iron levels were measured using commercially available kits as per the manufacturer's instructions (Crest Biosystem, Goa, India). Briefly, plasma (200 μl) was treated with an acidic buffer to allow iron bound to Transferrin to be released, along with reduction of $Fe^{3+}$ to $Fe^{2+}$, which upon reaction with Ferrozine formed a violet coloured complex, and absorbances were measured at 570 nm using Spectramax M2e (Molecular devices, Sunnyvale, CA, USA). The concentration of iron in plasma was calculated against a supplied standard (100 μg/dl).

Soluble CD163 was measured by ELISA; briefly, to anti-human CD163 coated 96-well plates, standards and samples (100 μl) were incubated at room temperature for 2.5 h. After 3 washes, biotinylated antihuman CD163 antibody (100 μl) was added for 1 h and using HRP-conjugated streptavidin (100 μl), binding was detected using TMB and absorbances measured at 450 nm; the concentration of sCD163 was calculated against a standard curve (0–8000 pg/ml).

## Immunophenotyping of peripheral blood monocytes

Peripheral blood was layered over a monocyte isolation medium (3:1; HiSep 1073, Himedia, Mumbai, India), and centrifuged (400$g$ for 30 minutes). The monocyte rich interface, after two washes in PBS, was resuspended in PBS and cell viability confirmed by trypan blue exclusion (>95%). Whole blood (100 μl) stained with anti-human CD14-Fluorescein isothiocyanate (FITC), transferrin receptor (CD71)-Allophycocyanin (APC), hemoglobin-haptoglobin receptor (CD163)-Phycoerythrin (PE) and hemopexin receptor (CD91)-PE was incubated for 20 minutes; erythrocytes were lysed with incubation in BD Fix-lyse lysing buffer for 10 minutes (2 ml), washed twice with PBS and finally resuspended in PBS (400 μl) for acquisition in a Flow Cytometer (BD FACSVerse). Monocytes were initially gated on their characteristic forward vs. side scatter, and quadrants set based on their fluorescence minus one (FMO) controls, after which on the basis of CD14 positivity 3000 cells were acquired per tube and analysed using BD FACS Suite software (BD Biosciences, San Jose, CA, USA). As the samples were obtained from field trips, there was a 24-48h lag period which adversely impacted on the cell morphology; accordingly, samples analyzed by Flow Cytometry were randomly selected, ensuring that at least 5–7 samples were analyzed per assay.

## Determination of intramonocytic labile iron pool

Monocytes ($5x10^5$/ml) were centrifuged (400$g$ X 5 minutes), resuspended in 500 μl PBS, and stained with Calcein acetoxymethyl ester (Calcein-AM, Molecular Probes, Carlsbad, CA, USA, 2.5 nM, 30 minutes, 37˚C) [17], after which fluorescence was acquired on a flow cytometer. Cells were morphologically gated, and 2000 monocytes were acquired per tube and analysed using BD FACS Suite software.

## Isolation of RNA and cDNA preparation

Total RNA from skin (4 mm punch biopsies) of patients with PKDL [n = 8; polymorphic (n = 4) and macular (n = 4)] or healthy individuals (n = 3, males undergoing voluntary circumcision) was isolated using the Trizol method and converted to cDNA using the cDNA Reverse Transcription Kit. Absolute quantification by droplet digital PCR was performed using gene specific primers for transferrin receptor (*TfR1/CD71*), haemoglobin/haptoglobin scavenging receptor (*CD163*), divalent metal transporter (*DMT-1/NRAMP-2*), Lipocalin-2 (*Lcn-2*), heme-oxygenase-1 (*HO-1*), *ferritin-H*, *NRAMP-1*, Ferroportin (*Fpn1*) and *Interleukin-10* (*IL-10*) (Table 1). Primers were designed from the database of National center for Biotechnology Information (NCBI) and specificity confirmed.

**Table 1. Primers and amplification conditions for RT-PCR (https://www.ncbi.nlm.nih.gov).**

| Name | Sequence (5'-3') | Annealing Temperature (°C) |
|---|---|---|
| *TfR1* (F) | CAGCAGAGACCAGCCCTTAG | 55.5 |
| *TfR1* (R) | TGCCTTGTGTGTTGTTTTCGT | |
| *CD163* (F) | ACCTCTTCAACAGACCCCCAGTGAA | 57.8 |
| *CD163* (R) | GAGGACTGAGAGCTCTTCTGGCATT | |
| *DMT-1* (F) | AGCCAGAGCCAGGTACTCAA | 57.8 |
| *DMT-1* (R) | GTGCAATGCAGGATTCAATG | |
| *Lipocalin-2* (F) | GGGAGAACCAAGGAGCTGAC | 56.5 |
| *Lipocalin-2* (R) | AGCTCCCTCAATGGTGTTCG | |
| *HO-1* (F) | TTCTCTCCCAACCCTGCTTGCGT | 57.8 |
| *HO-1* (R) | AGGTGGGCAGACCAAGGTTCAA | |
| *HIF-1α* (F) | ACCTATGACCTGCTTGGTGC | 59 |
| *HIF-1α* (R) | GGCTGTGTCGACTGAGGAAA | |
| *Ferritin* (F) | AATCCAAGACAGCCACACCTT | 60 |
| *Ferritin* (R) | TTGGGAAAGCTGCCCACTAA | |
| *NRAMP-1* (F) | AAACCCGGCCTGATTAAAGT | 57.8 |
| *NRAMP-1* (R) | GCCTGACGGAAAGAAGTG | |
| *Ferroportin* (F) | CGAGATGGATGGGTCTCCTA | 55.5 |
| *Ferroportin* (R) | ACCACATTTTCGACGTAGCC | |
| *IL-10* (F) | ACCCAGTCTGAGAACAGCTGC | 59.4 |
| *IL-`10* (R) | GTTCACATGCGCCTTGATGTCT | |

## Copy number analysis of the target genes using droplet digital PCR (ddPCR)

The cDNA from lesional or healthy skin (diluted with nuclease free water to a final concentration of 12.5 ng/μl) was used for ddPCR to quantify in terms of copy number. The reaction mixtures contained ddPCR EvaGreen Supermix (Bio-Rad Laboratories, Hercules, CA, USA), primers (6 μM) and template DNA (1 μl, 12.5 ng/μl) in 20 μl. Each reaction was then loaded into a sample well of an eight-well disposable cartridge (DG8), along with 70 μl of droplet generation oil. Droplets were formed using a QX200 droplet generator as per the manufacturer's instructions which were then transferred to a 96-well PCR plate to perform PCR (95°C for 5 min, followed by 40 cycles of 94°C for 30s and 60°C for 1 min, with a final extension at 98°C for 10 min). The annealing temperature and cycle number were optimised for incremental separation between positive and negative partitions in ddPCR. The resultant products were scanned on a QX200 Droplet Reader, and data analysed using QuantaSoft software. Any value of copy number/μl above 0.5 was considered as 1 copy, and if below 0.5, was considered as non-detectable. Values were expressed as copy number/20 μl.

## Detection of intracellular free ferric iron ($Fe^{3+}$) in lesional sites by Prussian blue staining

Formalin fixed paraffin embedded (FFPE) tissue sections were deparaffinised in xylene and rehydrated using descending grades of alcohol (100–70%) and distilled water. The slides were then incubated with a working solution of equal volumes (1:1) of 5% HCl (aq) and 5% Potassium ferrocyanide [$K_4Fe(CN)_6$] at room temperature for 20 minutes. After three washes with water, slides were counterstained using Nuclear Fast Red for 5 minutes and observed under a light microscope (EVOS FL Cell Imaging System, Waltham, MA, USA). Immunohistochemical analysis was performed for identification of macrophages using anti-human CD68 [18].

## Statistical analysis

Data was analyzed either by Mann-Whitney t test (in case of two groups) for non-parametric data and unpaired t test for parametric data using Graph Pad Prism software (version 5.0), $p < 0.05$ being significant. All data were expressed as median (Interquartile range or IQR) except the ddPCR data which was expressed as mean ± SEM (Standard Error of Mean). Correlation was by Pearson's correlation for parametric data and Spearman's rank correlation for non-parametric data, and the coefficient of correlation (r) when >0.4 was considered as relevant.

## Ethics statement

The study received approval from the Institutional Ethics Committee of School of Tropical Medicine, Kolkata and Institute of Post Graduate Medical Education and Research, Kolkata. Written informed consent was obtained from all individuals, and for a minor (<18 years), their legally accepted representative provided the same.

## Results

### Study population

Patients with PKDL (n = 25; **Table 2**) were randomly recruited by active or passive surveillance; the population included polymorphic (n = 10) and macular (n = 15) cases with a view to represent the present scenario of PKDL [12]. The rk39 test was positive in all cases while the presence of Leishman Donovan bodies was identified by Giemsa staining in all the polymorphic cases, but not in the macular variant. Additionally, the presence of parasites was confirmed by ITS-1 PCR [15]. Their hemoglobin levels and leukocyte counts were comparable with controls (**Table 2**). Amongst them, 21/25 patients (84%) gave a past history of VL and the median interval between VL and onset of PKDL was 5 years (**Table 2**). The parasite burden as quantified by qPCR was 5550 (1765–8891) parasites/µg of genomic DNA (**Table 2**).

### Increased entry of iron into circulating monocytes and lesional monocytes/macrophages in patients with PKDL

The entry of iron into monocytes/macrophages involves multiple ports of entry that include (a) the transferrin receptor TfR1 (CD71) that binds iron bound to ferritin and/or transferrin,

**Table 2. Study population.**

| Characteristics | Patients with PKDL (n = 25) | Healthy controls (n = 24) |
|---|---|---|
| Age (years)* | 29.00 (18.50–43.50) | 25.00 (7.70–29.50) |
| Sex ratio (M:F) | 2:3 | 1:1 |
| Hb (g/dl)* | 13.20 (11.60–14.20) | 13.23 (11.54–14.00) |
| WBC (count/mm³)* | 4.65 (3.80–5.40) | 5.35 (4.12–6.09) |
| Lesion type (Macular: Polymorphic) | 3:2 | NA |
| Disease duration (years)* | 2.00 (0.60–5.50) | NA |
| Interval between cure of VL and onset of PKDL (years)* | 5.00 (3.00–10.00) | NA |
| ITS-1 PCR +ve | 25/25, 100% | NA |
| Parasite load (number of parasites/µg of genomic DNA)* | 5550 (1765–8891) | NA |

*values are given as median (IQR); NA = Not applicable

(b) divalent metal transporter (DMT-1) for non-transferrin bound iron, or lipocalin-2 (Lcn-2) for siderophore bound iron that mediates transmembrane uptake of $Fe^{2+}$, (c) CD163 via phagocytosis of the hemoglobin-haptoglobin (Hb-Hp) complexes and CD91 for hemopexin bound heme uptake [19]. The frequency of TfR (CD71) positive $CD14^+$ monocytes was significantly raised by 2.42-fold *vis a vis* healthy controls, 16.46 (13.15–23.72) vs. 6.80 (5.31–7.84) %, p<0.001 (**Fig 1A**), but their frequency correlated poorly with the parasite load, r = 0.07.

CD163, the scavenger receptor for the haptoglobin-hemoglobin complex is expressed exclusively on monocytes/macrophages, and the frequency of $CD14^+163^+$ monocytes was significantly elevated by 6.74-fold in patients with PKDL as compared to healthy controls, 8.16 (5.59–9.61) vs. 1.21 (0.68–2.14) %, p<0.01 (**Fig 1B**), along with a significant 8.4 fold increase in the plasma levels of sCD163, 60154 (9519–77690) vs. 7046 (3094–13728) pg/ml. However, CD163 correlated poorly with parasite load, r = 0.15. Hemopexin (Hpx)-bound heme is scavenged by the CD91 receptor and the frequency of $CD14^+91^+$ monocytes remained unaltered in patients with PKDL as compared to healthy controls, 31.73 (29.85–33.62) vs. 30.87 (28.47–32.13)% (**S1 Fig**)

As PKDL is a dermal disease, the mRNA expression of *TfR1* and *CD163* were examined at the lesional sites in terms of copy number/20 μl in 4 polymorphic and 4 macular PKDL patients. In healthy controls, the mRNA expression of *TfR1* was undetectable, whereas in PKDL cases there was a substantial increase, being 8.43 ± 0.99 copies/20 μl (**Fig 1C**). Similarly, in case of *CD163*, a 4.07-fold higher copy number was detected in PKDL vs. healthy controls, 10.00 ± 1.63 vs. 2.46 ± 1.25 copies/20 μl, p<0.05, (**Fig 1D**). The mRNA expression of *CD163* correlated positively with parasite load, r = 0.58, but that was not so with *TfR1*, r = 0.14.

Another portal of entry for non-transferrin bound iron and $Fe^{2+}$ from endosomes is via the divalent metal transporter-1 (*DMT-1/NRAMP-2*). At the lesional sites, the mRNA expression of *DMT-1* was elevated, being 14.00 ± 3.24 copies/20 μl, whereas it was undetected in healthy controls (**Fig 1E**). Another approach for iron acquisition is via lipocalin-2 (*Lcn-2*) whose mRNA expression was increased by 7.29-fold as endorsed by absolute quantification in PKDL lesions being 12.17 ± 3.72 vs. 1.67 ± 0.07 copies/20 μl, p<0.05 (**Fig 1F**). However, both *DMT-1* and *Lcn-2* showed no correlation with parasite load, r = 0.22 and 0.11 respectively.

## Status of intracellular labile iron pool (LIP), iron recycling and storage in PKDL

In PKDL, as the circulating monocytes and dermal macrophages are alternatively activated [20], the labile iron pool could be raised. This was substantiated in PKDL (n = 8, polymorphic: macular = 1:1) by measuring the intra-monocytic labile iron pool (LIP) using Calcein-AM whose fluorescence is quenched by $Fe^{2+}$; accordingly, the fluorescence of Calcein is inversely proportional to the intracellular iron content [17]. The LIP was enhanced in patients with PKDL as compared to healthy controls, the fluorescence expressed as geometric mean fluorescence channel (GMFC) being 2.15-fold lower, 3378 (3039–4391) vs. 7259 (6741–8232), p<0.001 (**Fig 2A and 2B**); the LIP correlated positively with parasite load, r = 0.53. In PKDL as compared to healthy controls, the plasma iron levels were significantly lowered by 3.87-fold, being 52.30 (27.05–102.90) vs. 202.40 (146.20–265.70) μg/dl, p<0.001 and negatively correlated with the parasite load, r = -0.83, p<0.05. However, in contrast to VL, the hemoglobin (Hb) levels of patients with PKDL was comparable with healthy controls 13.20 (11.60–14.20) vs. 13.23 (11.54–14.00) g/dl.

In response to CD163-mediated heme uptake and possible iron overloading secondary to the enhanced presence of IL-10 [21], monocytes/macrophages counteract the pro-oxidant milieu via enhancement of the inducible heme-oxygenase-1(HO-1), which is downstream of

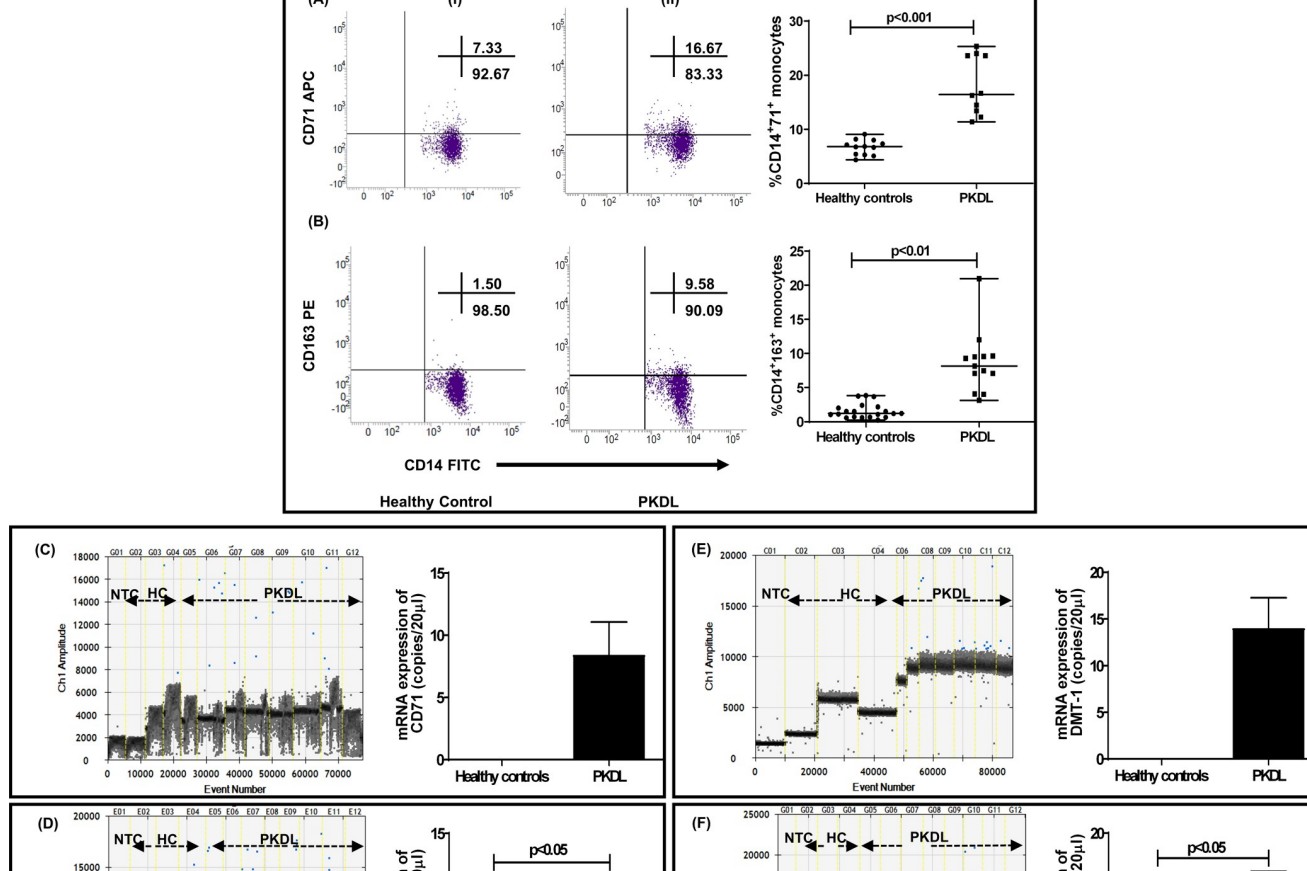

**Fig 1. Status of CD71, CD163 divalent metal transporter (DMT-1) and Lipocalin-2 in PKDL. A:** Representative profiles of the % of CD14+CD71+ circulating monocytes in a healthy control (**i**) and patient with PKDL (**ii**). Individual gates were set by using monocyte forward and side scatter characteristics and then with fluorochrome conjugated CD14-FITC. Scatter plots indicating frequency of CD71+ within CD14+ monocytes in healthy controls (n = 12, black filled circle) and patients with PKDL at presentation (n = 10, black filled square); each horizontal bar represents the median. **B:** Representative profiles of the % of CD14+CD163+ circulating monocytes in a healthy control (**i**) and patient with PKDL (**ii**) with individual gates being set using monocyte forward and side scatter characteristics followed by CD14-FITC. (**iii**) Scatter plots indicating frequency of CD163+ within CD14+ monocytes in healthy controls (n = 20, black filled circle) and patients with PKDL at presentation (n = 13, black filled square); each horizontal bar representing the median. **C&D:** One-dimensional plots of droplets measured for fluorescence signal (amplitude indicated on *y*-axis) emitted from the gene *CD71* (**C**) and *CD163* (**D**) at lesional sites. EvaGreen-bound positive droplets are shown in blue, while negative droplets are shown in black, along with bar graphs where data is expressed as mean ± SEM of the copy number/20 μl DNA. Bar graphs (open) denote healthy controls (n = 3) and filled bars represent patients with PKDL (n = 8). **E&F:** One-dimensional plots of droplets measured for fluorescence signal (amplitude indicated on *y*-axis) emitted from the gene *DMT-1* (**E**) and *lipocalin-2* (**F**) at lesional sites. EvaGreen-bound positive droplets are shown in blue while negative droplets are shown in black, along with bar graphs for data expressed as mean ± SEM of the copy number/20 μl DNA. Bar graphs (open) denote healthy controls (n = 3) while filled bars represent patients with PKDL (n = 8).

CD163 [22]. This was endorsed at the lesional sites, wherein the mRNA expression of *HO-1* was raised, 31.72 ± 13.12 copies/20 μl, whereas it was not detectable in healthy controls (**Fig 2C**), and positively correlated with parasite load, r = 0.83, p<0.05. A crucial role for mammalian oxygen sensing transcription factor hypoxia inducible factor-1α (HIF-1 α) has been

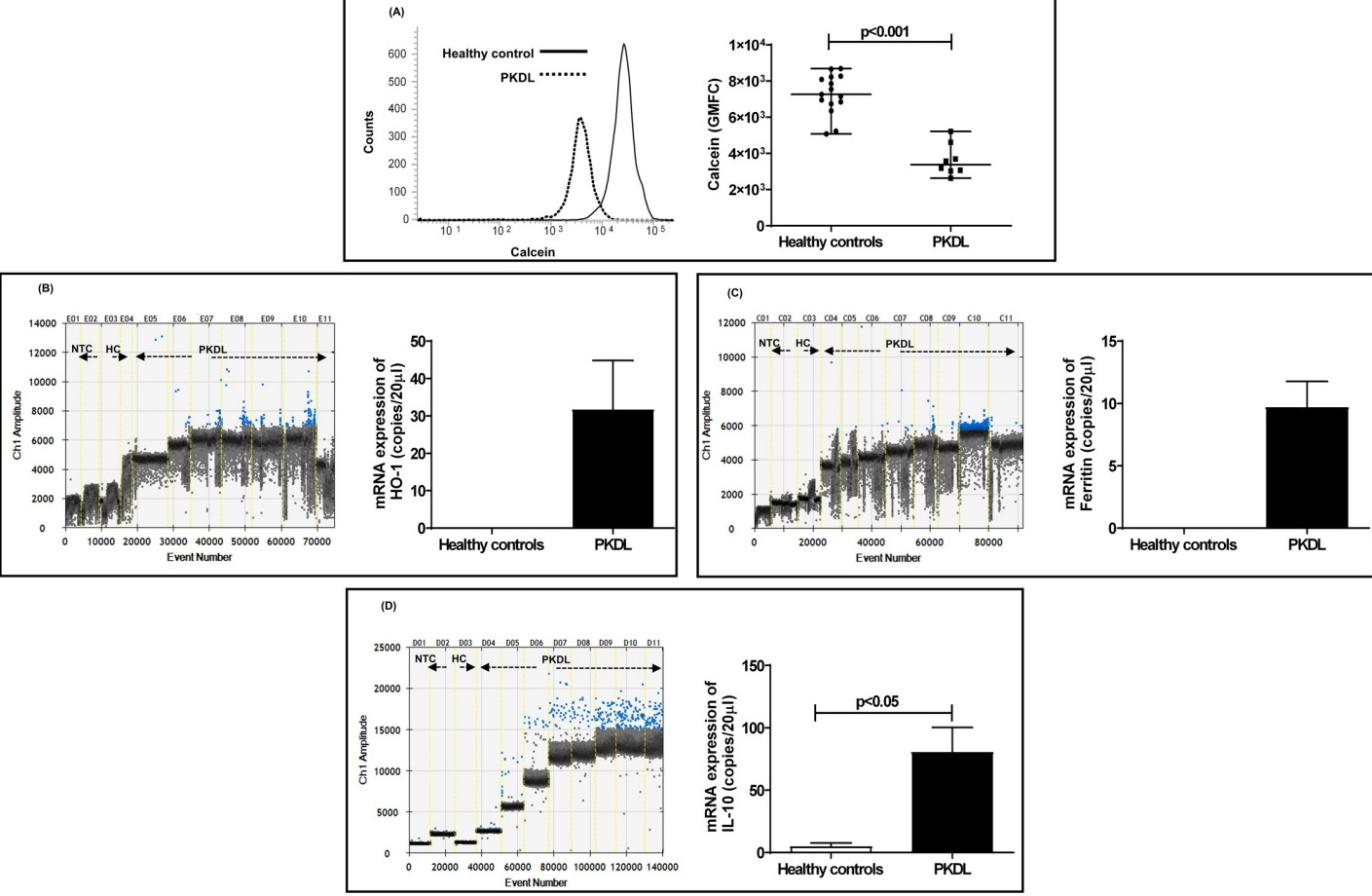

**Fig 2. Status of intramonocytic labile iron pool and heme-oxygenase-1 in PKDL. A:** Representative histogram profile of GMFC of calcein in circulating monocytes from a healthy control (solid line) and patient with PKDL (dotted line). Scatter plot for GMFC fluorescence of calcein representing the intracellular labile iron pool in circulating monocytes from healthy controls (n = 15, black filled circle) and patients with PKDL (n = 8, black filled square). Each horizontal bar represents the median. **B-D:** One-dimensional plots of droplets measured for fluorescence signal (amplitude indicated on *y*-axis) emitted from the gene *HO-1* (**B**), *Ferritin* (**C**) and *IL-10* (**D**) in lesional sites. EvaGreen-bound positive droplets are shown in blue while negative droplets are shown in black, along with bar graphs for data expressed as mean ± SEM of the copy number/20 µl DNA. Bar graphs (open) denote healthy controls (n = 3) while filled bars represent patients with PKDL (n = 7).

established in innate immunity against intracellular pathogens and its expression was measured. However, there was no alteration in the mRNA expression of *HIF-1α* in patients with PKDL as compared to healthy controls 2.34 ± 0.88 vs. 2.47 ± 1.24 copies/20 µl (**S2 Fig**).

Iron storage at the dermal sites was also examined in terms of the mRNA expression of the *ferritin* heavy chain, wherein healthy controls showed no detectable expression, but in PKDL cases, there was a substantial elevation being 9.70 ± 2.07 copies/20 µl (**Fig 2D**), and significantly correlated with parasite load r = 0.75, p<0.05. This was accompanied by a significant 16.44-fold increase in the *IL-10* copy number, 80.57 ± 19.71 vs. 4.90 ± 2.75 copies /20 µl, p<0.05 (**Fig 2E**). *CD163* and *HO-1* positively correlated with *IL-10*, r = 0.42 and 0.64 respectively.

## Increased mRNA expression of iron exporters at the dermal sites in patients with PKDL

The natural resistance macrophage protein1 (NRAMP-1or Slc11a1) has been characterized as a late phagosomal protein that exports iron from phagosomes to the cytoplasm, followed by

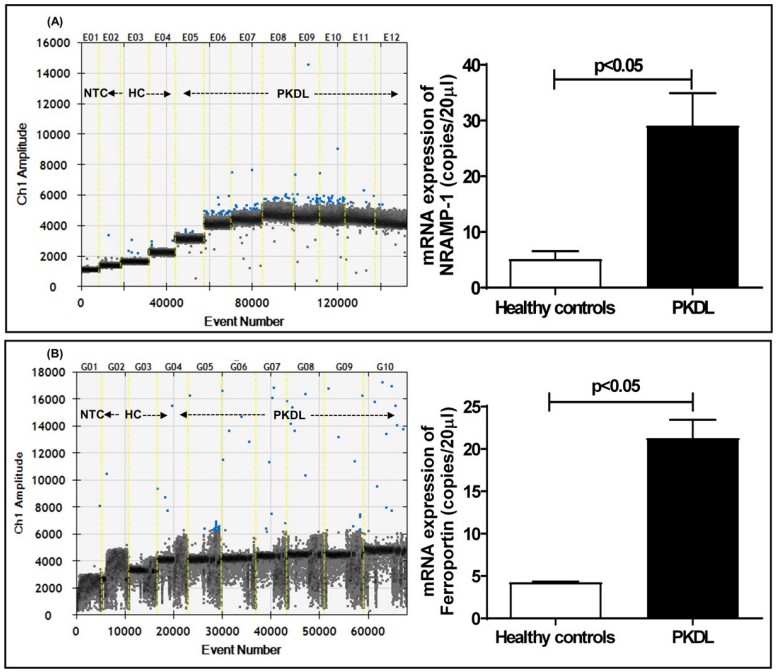

**Fig 3. mRNA expression of NRAMP-1 and Ferroportin in dermal lesions of PKDL. A&B:** One-dimensional plots of droplets measured for fluorescence signal (amplitude indicated on *y*-axis) emitted from the gene *NRAMP-1* (**A**) and *Ferroportin* (**B**) in lesional sites. EvaGreen-bound positive droplets are shown in blue while negative droplets are shown in black, along with bar graphs for data expressed as mean ± SEM of the copy number/20 µl DNA in lesional sites. Bar graphs (open) denote healthy controls (n = 3) while filled bars represent patients with PKDL (n = 8 for *NRAMP-1* and n = 7 for *Ferroportin*).

iron being subsequently exported through ferroportin present on the cell membrane. In PKDL, the lesional mRNA expression of *NRAMP-1* was significantly enhanced by 5.67-fold as compared to healthy controls, 29.07 ± 5.85 vs. 5.13 ± 1.43 copies/20 µl, $p < 0.05$, (Fig 3A), but correlated marginally with the parasite load, r = 0.39. At the lesional sites, the mRNA expression of ferroportin significantly increased by 4.95-fold in PKDL, 21.29 ± 2.17 vs. 4.30 ± 0.10 copies/20 µl, $p < 0.05$ (Fig 3B), and positively correlated with the parasite load, r = 0.86, $p < 0.05$. Despite the substantial infiltration of CD68[+] macrophages in the lesional sites, staining for Prussian blue which represents the proportion of $Fe^{3+}$, was negative (**S3 Fig**).

## Discussion

The acquisition of iron by intracellular pathogens like *Leishmania* parasites is analogous to a double edged sword, as iron is essential for their cellular metabolic processes and pathogenicity [5], being a component of several iron-dependent proteins [e.g. iron superoxide dismutase (FeSOD), ascorbate peroxidase, cytochrome b5 (CytB5) and cytochrome p450 (CYP)] that participate in detoxification of reactive oxygen species [23–25]. Additionally, as iron is a component of ribonucleotide reductase and iron clusters present in the mitochondrial respiratory chain, it is critical for DNA synthesis and energy metabolism respectively [26]. However, this very property of electron transfer makes iron potentially dangerous via its participation in oxidation/reduction reactions, where by donating electrons to $O_2$ and $H_2O_2$, it can generate toxic molecules like superoxide anion and hydroxyl radicals [27]. Accordingly, for *Leishmania* to survive and replicate within a hostile iron limiting parasitophorous vacuole (PV), the cellular uptake, distribution, storage and export of iron is a tightly regulated process [6].

The delivery of iron to cells is accomplished by multiple approaches that include its complexation with transferrin (Tf), a monomeric plasma iron-binding protein, which upon binding to cell-surface receptors (transferrin receptor or TfR1) facilitates the entry of iron via endocytosis [28]. However, as the $Fe^{3+}$ released from transferrin is essentially insoluble at physiological pH, it is reduced to $Fe^{2+}$ within endosomes by ferric reductase (STEAP3), and eventually contributes towards the labile iron pool (LIP) [28]. In the peripheral blood of patients with PKDL, the uptake of iron was supported by an increased frequency of CD71/TfR1 (**Fig 1A**) and was corroborated by its enhanced mRNA expression at lesional sites (**Fig 1C**). *Leishmania* upon scavenging iron from LIP causes activation of iron-sensing proteins, and the resultant enhanced interaction of Iron Responsive Elements (IREs) with TfR1 translates into an increased uptake of iron [29]. Additionally, as patients with PKDL demonstrate a mixed Th1-Th2 cytokine milieu with a tilt towards Th2, the increased levels of IL4 and IL-10 [30] too can facilitate an increased mRNA expression of *TfR1* [31]. However, this did not occur in patients with VL, as a decreased expression of *TfR1* was reported [31]. Similarly, studies regarding *Legionella* also reported decreased levels of TfR1 in activated macrophages which were attributed to the IFN-γ mediated pro-inflammatory environment [32]. Another intracellular pathogen that enhances its iron pool via increased entry through TfR1 is *Francisella tularensis* [33], whereas *Salmonella typhimurium*, *Legionella pneumophila* and *Neisseria spp* sequester iron as transferrin via siderophores [34].

While the presence of iron in endosomal membranes is associated with entry via transferrin, the uptake of non-transferrin bound iron is mediated by the membrane bound divalent metal transporter 1 (DMT1), also known as NRAMP-2 or solute carrier family 11 member 2, SLC11A2 [35]. NRAMP-2 is a symporter of $H^+$ and metal ions, and under physiological conditions is localized in the early endosomal membranes where it is responsible for delivery of extracellularly acquired bivalent cations into the cytosol [36]. Intracellular organisms like *Francisella tularensis* upregulate DMT-1 expression [37] and similarly, at lesional sites of PKDL cases, the increased mRNA expression of *DMT-1* endorsed its possible utilization by parasites to enhance influx of iron (**Fig 1E**). However, in VL, an unaltered expression of *DMT-1* was observed, and could be attributed to it being an acute disease *vis-a-vis* the chronicity of PKDL [31].

To counteract siderophore mediated iron acquisition by pathogens, host monocytes infected with *Mycobacterium tuberculosis* produce Siderocalin, or lipocalin 2 (Lcn-2), and this translated into attenuation of infection [38, 39]. Although evidence regarding the role of siderophores and/or siderophore receptors in acquisition of iron by *Leishmania* is limited, in PKDL, an increased lesional mRNA expression of *lcn-2* was demonstrated (**Fig 1F**), attributable to a possible pleiotropic effect of cytokines. *In-vivo* studies have shown that following LPS administration to neutrophils and macrophages, the resultant activation of the toll-like receptor (TLR)-4 leads to increased secretion of Lcn-2. Additionally, induction of Lcn2 can be triggered by other TLR ligands and cytokines, including IL-1β, IL-6, IL-10, IL-17, IL-22, and TNF-α [40]. Accordingly, it may be proposed that in PKDL cases, the increased expression of *Lcn-2* may be attributed to the increased lesional mRNA expression of *IL-10* (**Fig 2E**), along with raised circulating levels of pro- (TNF-α, IL-6, IL-1β and IL-8) and anti-inflammatory (IL-4, IL-10, IL-13 and TGF-β) cytokines, the latter being signature molecules of M2 polarization [20, 30]. The SLC39/ZIP family transporters are a new class of iron-trafficking proteins containing 14 members which includes ZIP8 and ZIP14 that transport non-transferrin bound iron above pH 7 and near physiological pH, whereas DMT-1 is most efficient at pH 5.5, which corresponds to the pH of acidified endosomes [41]. Therefore, the impact of these newer transporters on *Leishmania*-macrophage interaction should be explored and may well explain some of the apparent paradoxes observed.

Besides TfR1, macrophages express the hemopexin receptor (CD91), which takes up heme bound to the heme-sequestering protein hemopexin [42]. However, in peripheral blood of PKDL cases, the frequency of CD91 remained unaltered. A similar function is attributed to CD163, another entry point for iron through which heme bound iron is scavenged in the form of hemoglobin bound to haptoglobin [42]. CD163 is considered as a hallmark of alternatively activated M2 macrophages, a phenotype that sustains the disease chronicity in PKDL [20, 43]. In peripheral blood, the enhanced frequency of circulatory CD163 (**Fig 1B**) was endorsed by its increased mRNA expression at lesional sites (**Fig 1D**). This enhanced uptake of heme-bound iron can induce hemeoxygenase-1 (HO-1), an enzyme responsible for degrading heme to biliverdin, carbon monoxide and $Fe^{2+}$, as also mediate attenuation of the immunopathology caused by oxidative stress [44]. During *Leishmania* infection, the host tightly regulates the soluble $Fe^{2+}$ by deploying multiple options that includes (i) incorporation into the LIP, (ii) storage as ferritin, (iii) export via ferroportin [Fpn, 45] and/or (iv) entry into the phagolysosome. The increased mRNA expression of *HO-1* has been implicated in the pathogenesis of several infectious diseases, as its activation following heme degradation translates into decreased formation of the heme containing NADPH-oxidase complex. This leads to an attenuated generation of reactive oxygen species (ROS) and provides for a pro-parasitic environment [46]. Indeed, the replication of *Leishmania chagasi* was favoured by induction of HO-1, as it dampened leishmanicidal mechanisms such as generation of nitric oxide and ROS [47]. Furthermore, in PKDL the HO-1 catalyzed generation of CO, can lead to decreased generation of superoxide, and along with an increased availability of reduced glutathione, facilitate creation of an anti-oxidant, pro-parasitic milieu [20, 48]. In PKDL cases, the lesional mRNA expression of *HO-1* was substantially elevated (**Fig 2C**) and correlated positively with the parasite load. Another factor that enhances HO-1 activation is IL-10 [49] whose upregulation (**Fig 2E**) represents an additional subversive mechanism adopted by *Leishmania* to escape the oxidative burst, and therefore its manipulation could well be considered as a therapeutic target.

Intracellular pathogens can access the LIP, the metabolically active fraction of cytosolic iron that is loosely bound to low molecular-weight chelators and represents the free cytoplasmic iron pool [19, 50]. As LIP is the transition zone between import, cellular utilization and storage, it responds to the cell's metabolic needs. During infection with *Francisella tularensis*, there is an active $Fe^{2+}$ acquisition system associated with a sustained increase in the LIP [51]. Similarly, erythrocytes infected with *Plasmodium falciparum* demonstrated an increased LIP that was associated with parasite maturation [52]. Similarly, in PKDL, the increased LIP, (**Fig 2A and 2B**) via its ability to inhibit binding of the transcription repressor Bach1 to specific sequences in the HO-1 promoter, could potentially increase the expression of HO-1 and enhance parasite survival [53]. Although the calcein-AM (calcein-acetoxymethyl ester) method is a widely used technique to measure the LIP, it can only measure the cytoplasmic LIP, as it cannot enter cellular compartments. Therefore, the changes, if any, in the low-mass iron present in the phagolysosomal compartment will get overlooked. This is particularly relevant in Leishmaniasis as amastigotes reside within the phagolysosome, whereas this compartment, owing to its low pH, cannot be accessed by Calcein. Therefore, there could well be an under representation of the iron present within the amastigotes [54]. As an excess of free $Fe^{2+}$, secondary to its enhanced entry (**Fig 1**), can be potentially hazardous owing to its ability to support the generation of free radicals, an increased LIP is generally associated with an upregulation of Ferritin-H, which along with Ferritin-L subunits, forms a multimeric protein complex and stores $Fe^{2+}$ [55]. This was possibly adopted by PKDL cases as evident by the upregulation of lesional *ferritin* (**Fig 2D**), although it was not in concordance with the classical ferritin[low] phenotype associated with alternatively activated M2 polarized macrophages [22].

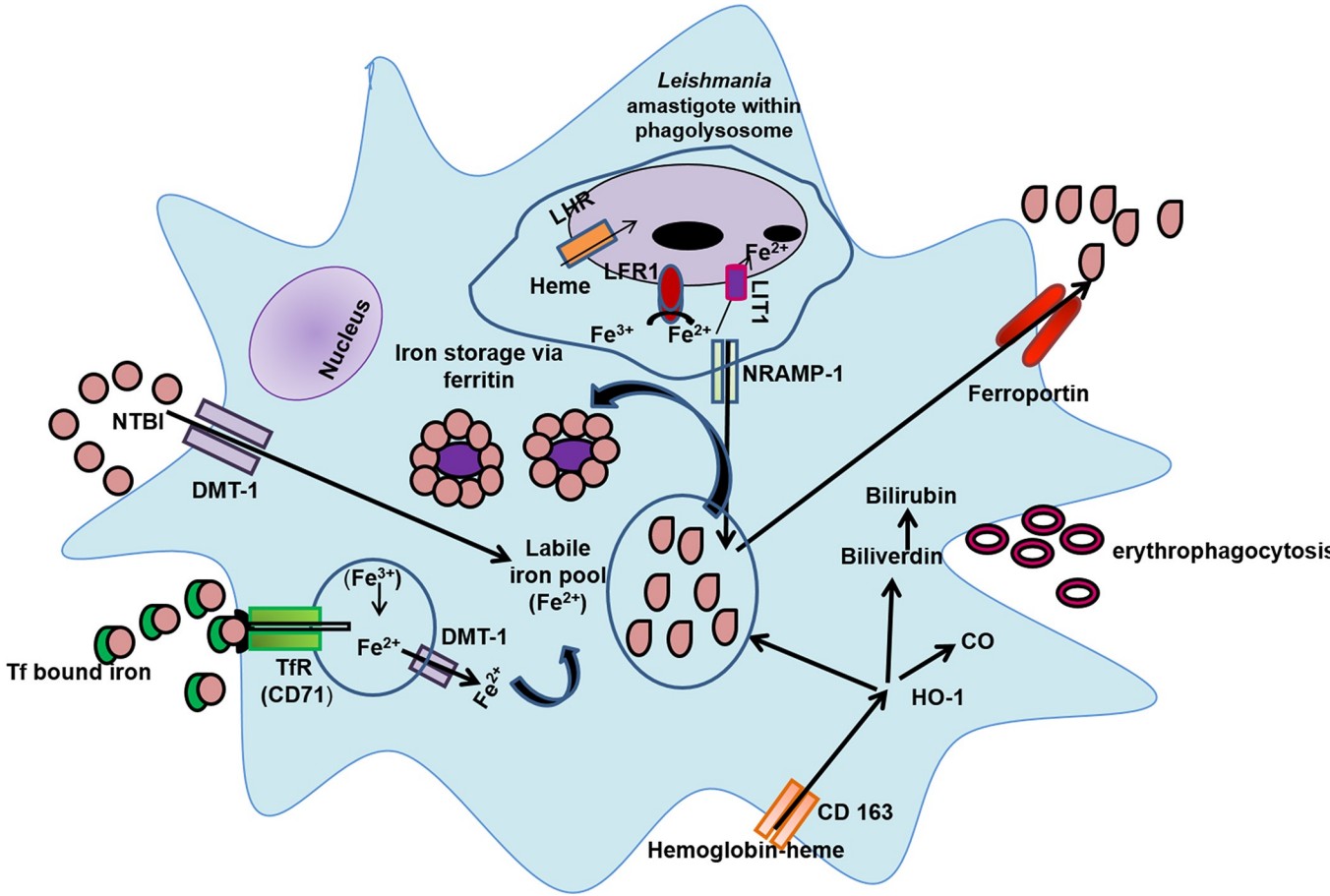

**Fig 4. Iron trafficking in *Leishmania* infected monocytes/macrophages.** Following the phagocytosis of senescent erythrocytes (erythophagocytosis), the heme-bound iron is released and recycled to the labile iron pool. Other alternatives for uptake of iron include Transferrin-bound iron that is internalized through the transferrin receptor (CD71, TfR1), non-transferrin bound iron (NTBI) is taken up through the divalent metal transporter-1 (DMT-1), and hemoglobin-bound heme through CD163. Following metabolism of heme by hemeoxygenase-1 (HO-1), there is generation of carbon monoxide (CO), bilirubin and free iron, the latter being stored as ferritin or is added to the labile iron pool, and/or can be exported via ferroportin. *Leishmania* amastigotes can augment the iron-poor environment of the parasitophorous vacuole (PV) via *Leishmania* ferric reductase-1 (LFR1) which converts $Fe^{3+}$ to $Fe^{2+}$, which is then translocated across the membrane by *Leishmania* $Fe^{2+}$ iron transporter (LIT1). Additionally, monocytes-macrophages strive to limit the availability of iron to intracellular pathogens by enhancing its removal from the parasitophorous vacuole (PV) to the cytoplasm via natural resistance-associated macrophage protein (NRAMP-1), which is further exported through ferroportin.

The replication of *Leishmania* within acidified, hydrolase-rich microbicidal phagolysosomes requires iron and their $Fe^{2+}$ acquisition strategies includes ferric reductase (LFR1), which by converting Ferritin bound $Fe^{3+}$ to $Fe^{2+}$ facilitated the entry of *L. amazonensis* amastigotes via the *Leishmania* $Fe^{2+}$ transporter 1(LIT-1) [50]. Alongside, macrophages express a metal transporter NRAMP-1 that functions as a pH dependent divalent cation efflux pump whose crucial role is modulating the access of pathogens to iron as substantiated by studies in NRAMP $^{-/-}$ animals [2]. It is a highly pH dependent antiporter that fluxes metal ions in either direction against a proton gradient. Furthermore, as NRAMP-1 is localized in late endosomal/lysosomal membranes, it delivers bivalent cations from the cytosol into this acidic compartment and therefore can influence antimicrobial activity [36]. Accordingly, the increased transcriptional expression of *NRAMP-1* in PKDL (Fig 3A) could be attributed to the subtle iron homeostasis achieved by parasites within phagolysosomes.

Ferroportin (Fpn) is an established mammalian iron exporter whose regulation is critical for iron homeostasis, and its alterations can translate into iron deficiency or iron overload. It has been proposed that to increase the availability of iron, host cell derived hepcidin promotes the degradation of Fpn which translates into increased availability of iron during infections with *L. amazonensis* [56, 57], and has been substantiated in patients with VL [31]. However, in *L. donovani* models of infection where despite the levels of Hepcidin being raised, the mRNA expression of *Fpn* too was elevated [57]. Similarly, in PKDL cases an increased mRNA expression of *Fpn* was demonstrated (Fig 3B); however it should be confirmed at a translational level. Similarly, in intracellular pathogens, *Salmonella* and *Mycobacterium*, an enhanced expression of Fpn has been reported, that could be attributed to either the host and/or the pathogen regulating the export of iron to limit its availability to the installed pathogen [58–60].

Taken together, this study has established that monocytes/macrophages sourced from patients with PKDL have several ports of entry that includes TfR1, DMT-1 and lipocalin-2 (Fig 4). Additionally, there is an increased uptake of heme bound iron by CD163. This results in an increased availability of HO-1, which in turn increased the presence of $Fe^{2+}$ resulting in an enhancement of the intramonocytic labile iron pool and ferritin, thus providing the pathogen with an adequate source of iron, which can be sequestered as necessary into phagolysosomes by LIT-1 (Fig 4). However, in view of iron being likened to a 'double edged sword', it is necessary to prevent iron mediated toxicity, and therefore host macrophages ensure this via an adequate efflux of iron through the phagolysosomal exporter NRAMP-1 and the cellular exporter Fpn (Fig 4). Collectively, to eliminate an intracellular pathogen like *Leishmania*, iron deprivation is necessary and is potentially achievable by either restricting the availability of iron i.e. nutritional immunity, or by dampening the microbial iron transporters, and both approaches could serve as novel therapeutic interventions.

## Supporting information

**S1 Checklist. STROBE checklist.**
(DOCX)

**S1 Fig. Representative profiles of the % of CD14$^+$CD91$^+$ circulating monocytes in a healthy control (i) and patient with PKDL (ii).** Individual gates were set by using monocyte forward and side scatter characteristics and then with fluorochrome conjugated CD14-FITC; **(iii)** scatter plots indicating frequency of CD91$^+$ within CD14$^+$ monocytes in healthy controls (n = 6, black filled circle) and patients with PKDL at presentation (n = 5, black filled square);each horizontal bar represents the median.
(TIF)

**S2 Fig. One-dimensional plots of droplets measured for fluorescence signal (amplitude indicated on y-axis) emitted from the gene *HIF-1α* in lesional sites.** EvaGreen-bound positive droplets are shown in blue while negative droplets are shown in black, along with bar graphs for data expressed as mean ± SEM of the copy number/20 μl DNA Bar graphs (open) denote healthy controls (n = 2) while filled bars represent patients with PKDL (n = 8).
(TIF)

**S3 Fig. Status of intracellular free Fe$^{3+}$ in lesional biopsies. A&B:** Representative H&E profiles from dermal biopsies of a healthy control and patient with PKDL (magnification 10X). **C&D:** Representative immunohistochemical profiles of CD68$^+$ macrophages from dermal biopsies from a healthy control and patient with PKDL (magnification 10X). **E&F:** Representative Prussian blue stained profiles from dermal biopsies of a healthy control and patient with PKDL (magnification 10X) showing absence of free ferric ion. **G&H:** Positive control

(Hemochromatosis liver section; magnification 10X and 40X).
(TIF)

## Author Contributions

**Conceptualization:** Aishwarya Dighal, Debanjan Mukhopadhyay, Mitali Chatterjee.

**Data curation:** Aishwarya Dighal, Debanjan Mukhopadhyay, Ritika Sengupta, Srija Moulik, Surya Jyati Chaudhuri, Nilay K. Das, Mitali Chatterjee.

**Formal analysis:** Aishwarya Dighal, Debanjan Mukhopadhyay, Ritika Sengupta, Shibabrata Mukherjee, Susmita Roy, Mitali Chatterjee.

**Funding acquisition:** Mitali Chatterjee.

**Investigation:** Aishwarya Dighal, Debanjan Mukhopadhyay, Shibabrata Mukherjee, Surya Jyati Chaudhuri, Nilay K. Das, Mitali Chatterjee.

**Methodology:** Aishwarya Dighal, Debanjan Mukhopadhyay, Ritika Sengupta, Srija Moulik, Shibabrata Mukherjee, Susmita Roy.

**Project administration:** Mitali Chatterjee.

**Resources:** Surya Jyati Chaudhuri, Nilay K. Das, Mitali Chatterjee.

**Software:** Aishwarya Dighal, Debanjan Mukhopadhyay, Shibabrata Mukherjee, Mitali Chatterjee.

**Supervision:** Mitali Chatterjee.

**Validation:** Aishwarya Dighal, Debanjan Mukhopadhyay, Srija Moulik, Susmita Roy, Mitali Chatterjee.

**Visualization:** Aishwarya Dighal, Debanjan Mukhopadhyay, Mitali Chatterjee.

**Writing – original draft:** Aishwarya Dighal, Debanjan Mukhopadhyay, Mitali Chatterjee.

**Writing – review & editing:** Aishwarya Dighal, Debanjan Mukhopadhyay, Mitali Chatterjee.

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
