## [Decision Letter · Decision Letter 0]

15 Sep 2019

Dear Prof. Chatterjee:

Thank you very much for submitting your manuscript "Iron trafficking in patients with Indian Post kala-azar dermal leishmaniasis" (PNTD-D-19-01246) for review by PLOS Neglected Tropical Diseases. Your manuscript was fully evaluated at the editorial level and by independent peer reviewers. The reviewers appreciated the attention to an important topic but identified some aspects of the manuscript that should be improved.

We therefore ask you to modify the manuscript according to the review recommendations before we can consider your manuscript for acceptance. Your revisions should address the specific points made by each reviewer.

(1) A letter containing a detailed list of your responses to the review comments and a description of the changes you have made in the manuscript.

(2) Two versions of the manuscript: one with either highlights or tracked changes denoting where the text has been changed (uploaded as a "Revised Article with Changes Highlighted" file ); the other a clean version (uploaded as the article file).

(3) If available, a striking still image (a new image if one is available or an existing one from within your manuscript). If your manuscript is accepted for publication, this image may be featured on our website. Images should ideally be high resolution, eye-catching, single panel images; where one is available, please use 'add file' at the time of resubmission and select 'striking image' as the file type. 

Please provide a short caption, including credits, uploaded as a separate "Other" file. If your image is from someone other than yourself, please ensure that the artist has read and agreed to the terms and conditions of the Creative Commons Attribution License at http://journals.plos.org/plosntds/s/content-license (NOTE: we cannot publish copyrighted images). 

(4) Appropriate Figure Files 

Please remove all name and figure # text from your figure files upon submitting your revision. Please also take this time to check that your figures are of high resolution, which will improve both the editorial review process and help expedite your manuscript's publication should it be accepted. Please note that figures must have been originally created at 300dpi or higher. Do not manually increase the resolution of your files. For instructions on how to properly obtain high quality images, please review our Figure Guidelines, with examples at: http://journals.plos.org/plosntds/s/figures

While revising your submission, please upload your figure files to the Preflight Analysis and Conversion Engine (PACE) digital diagnostic tool, https://pacev2.apexcovantage.com/ PACE helps ensure that figures meet PLOS requirements. To use PACE, you must first register as a user. Then, login and navigate to the UPLOAD tab, where you will find detailed instructions on how to use the tool. If you encounter any issues or have any questions when using PACE, please email us at figures@plos.org.

We hope to receive your revised manuscript by Nov 14 2019 11:59PM. If you anticipate any delay in its return, we ask that you let us know the expected resubmission date by replying to this email.

To submit your revised files, please log in to https://www.editorialmanager.com/pntd/

Sincerely,

Syamal Roy

Guest Editor

Brian Weiss

Deputy Editor

The authors should address the concerns raised by the reviewes. It is an important piece of contribution.

The authors have claimed that iron metabolism alone is a major mechanism of parasite adaptation. I think they should tone down such a conclusion.

Reviewer's Responses to Questions

**Key Review Criteria Required for Acceptance?**

**Methods**

-Are the objectives of the study clearly articulated with a clear testable hypothesis stated?

-Is the study design appropriate to address the stated objectives?

-Is the population clearly described and appropriate for the hypothesis being tested?

-Is the sample size sufficient to ensure adequate power to address the hypothesis being tested?

-Were correct statistical analysis used to support conclusions?

-Are there concerns about ethical or regulatory requirements being met?

Reviewer #1: Major revision

Reviewer #2: There is no additional experiments to be done. The data are robust and aligned with the study objective. One aspect on statistics that deserves better explanation are concerned with the threshold to define significan and non significan correlation between parasite load and expressed biomarkers.

Reviewer #3: Are the objectives of the study clearly articulated with a clear testable hypothesis? Yes

Is the study design appropriate to meet the stated objectives? Is correct

Is the population clearly described and appropriate for the hypothesis being tested? Yes

Is the sample size sufficient to ensure adequate energy to deal with the hypothesis being tested? Yes

- Was the correct statistical analysis used to support the conclusions? It's correct

Are there concerns about ethical or regulatory requirements being met? No worries, the work was submitted to the ethics committee.

I suggest placing reference on the primers used.

**Results**

-Does the analysis presented match the analysis plan?

-Are the results clearly and completely presented?

-Are the figures (Tables, Images) of sufficient quality for clarity?

Reviewer #1: (No Response)

Reviewer #2: Quality presentation of figures need to be improved, not in terms of scientific meaning but like art graphs.

Reviewer #3: Does the analysis presented match the analysis plan? Yes

-Are the results clearly and completely presented? Yes

-Are the figures (Tables, Images) of sufficient quality for clarity? The figures are not clear, need to improve the quality.

**Conclusions**

-Are the conclusions supported by the data presented?

-Are the limitations of analysis clearly described?

-Do the authors discuss how these data can be helpful to advance our understanding of the topic under study?

-Is public health relevance addressed?

Reviewer #1: (No Response)

Reviewer #2: This work contributes to PKDL understanding which is a positive and wellcomed contribution. The conclusions are supported by presented data but there are other possible interpretations that cannot be discarded. The literature related to the study subject permits a more critical discussion and the authors should point out the inherent limitations. This will not decrease the value of the research work but will expose the methodological limitations imposed by the current technology.

Reviewer #3: Are the conclusions supported by the data presented?

-Are the limitations of analysis clearly described? Yes

-Do the authors discuss how these data can be helpful to advance our understanding of the topic under study? Yes

-Is public health relevance addressed? Yes

**Editorial and Data Presentation Modifications?**

Reviewer #1: (No Response)

Reviewer #2: As it is mentioned in the Summary and general comments, the writing requires to be re-structured because some aspects of the parasite iron regulation are put at the same relevance category than the host. This work is focused on the host ant parasite participation must be quoted to stress their effect on iron host regulation.

Reviewer #3: (No Response)

**Summary and General Comments**

Reviewer #1: (No Response)

Reviewer #2: The PNTD-D-19-01246 entitled “Iron trafficking in patients with Indian Post kala-azar dermal leishmaniasis” analyzed different macrophage biomarkers involved in iron homeostasis that may play a role in host-parasite interactions. The manuscript (MS) results are addressed to fill a major gap of knowledge on those mechanisms that Leishmania intracellular amastigotes have to survive within host macrophages.

The main author´s claim is that iron influx gateways are overexpressed and as consequence iron will be available to intracellular amastigotes and therefore they will promote Leishmania infection persistence. On the other hand, iron macrophage exporters NRAMP-1 and ferroportin mRNA, although overexpressed, would not be enough to impair Leishmania survival.

The research carried out is original because there are no reports on this subject and it is a key topic to understand Leishmania intracellular survival and evading host defense mechanisms occurring in PKDL patients. Moreover, if you search for similar topic for the classic visceral leishmaniasis there are scarce number of similar approaches. 

MS reported differences on iron homeostasis between PKDL patients and healthy volunteers are of such magnitude that authors´ main conclusions provides robust experimental support to propose iron metabolism as a major topic for parasite adaptation. Nevertheless, there are MS structure and specific aspects that must be addressed by MS authors:

MS structure:

The Discussion is the MS section that demands more attention to ensure fluid reading. For example, at second paragraph where the authors start to discuss the iron entry to macrophages (line 322 to 336), dealing with observed TfR1 increase expression in PKDL patients, this is perturbed by referring the parasite iron acquisition pathways, like the ferrous iron transporter LIT-1 that scavenges iron from the labile iron pool (LIP) (LIT-1 was not mentioned by its name but quoted in reference 31). This paragraph should be only centered in TfR1, even more because the authors have identified a clear difference between PKDL and TfR1 reported observation for VL (quoted in ref 32). If they want to mention parasite iron acquisition it should be like a possible explanation to observed macrophage TfR1 increase. The MS is not on parasite iron incorporation but on iron homeostasis perturbation in PKDL patients. The authors should extend further the comparison with VL and other diseases caused by intracellular pathogens. Here, I would translate the paragraph on Ferroportin (Fpn) (lines 390 to 395), where Fpn is upregulated in PKDL (based upon mRNA levels) whereas in VL is downregulated via Ferroportin degradation.

The third paragraph, where the increase of DMT-1 and lipocalin 2 are mentioned, there is no discussion that contributes to either define apparent contradictions or to explain them. If lipocalin 2 is increased in Mycobacterium infected cells to counteract bacterial siderophores, as mentioned by authors, nothing is mentioned to the fact that there are no siderophores described for Leishmania. If this lipocalin 2 increase has another possible explanation I would have discussed other factors, for example IL-10 that was also found upregulated in PKDL patients. This apparent paradox is much more interesting because there is an interesting concept where extracellular pathogens (at least for bacteria) evokes iron sequestration by macrophages whereas intracellular ones go in opposite direction, i.e. invaded host cells exports iron to limit its availability to installed pathogens (Gan Z. et al. Metallomics, 2019. 11: 454- 461). Again, the indirect effects of other factors like cytokines or HO-1 related signals might be involved.

The Figure texts are immersed within different parts of the main body text. This fact affects fluid reading.

I would re-write paragraphs 404 to 407 to make explicit that the observed measurements may reflect the “double edged sword” mentioned by the MS authors at the beginning of discussion section.

The authors should also mention that not all described transporters have been studied. See the review of Bogdan A.R. et al. Trends in Biochemical Sciences 2016. 41: 274 -286, where other transporters are mentioned and therefore some of the observed apparent paradoxes might be explained by their participation in Leishmania macrophage infection.

I would expect to discuss the role of pH on transporter or iron binding molecules. One good example of this is the work of Goswami T. et al. Biochem. J. (2001) 354, 511–519, where they demosntrate that Nramp2 is a H+/bivalent catión antiporter where the flow of ions depends on its localization an affected by the pH. 

Specific Comments 

The use of Calcein acetoxymethyl ester to measure labile iron pool (LIP) has limitations to measure non bound iron when present inside membrane compartments with acidic pH (Tenopoulo M. et al. Biochem. J. (2007) 403, 261–266; doi:10.1042/BJ20061840). Because amastigotes live within the phagolyzosome, a compartment that cannot be reached by Calcein and/or the acidic pH, it will cause an error when trying to interpret the iron impact on amastigotes. The measures iron should correspond to cytoplasmic iron and therefore without direct impact on amastigotes that are in another compartment.

To mention enhanced entry of siderophore bound iron (lines 397 to 399) should be conciliated with the fact that Leishmania does not have reported siderophores. Perhaps a pleiotropic effect of cytokines would be an approach to this fact.

The sentence comprised in lines 387 to 388 may have several interpretations according the readers. I suggest to re-write to understand what do the authors wants to express.

It is incorrect to say in lane 78 that the role of iron in leishmaniasis was derived from in vitro models. There are in vivo models and also indirect results working with patients where iron metabolism has been addressed.

The Figures requires a better art graph but more important is to review if they are properly presented. For example in Figure 1 A, i) and ii) each of them has a different gateway that alters the values at such extent that comparison conditions are not adequate. In contrast Figure 1 B is correct. Like this, there are minor but important revision an corrections to be made. Another example is in lane 214 to lane 216 wehere the ratio does not correspond to the quoted values but in contrast lanse 218 to 220 lead to a ratio with the expected value.

Reviewer #3: (No Response)

PLOS authors have the option to publish the peer review history of their article (what does this mean?). If published, this will include your full peer review and any attached files.

Reviewer #1: No

Reviewer #2: No

Reviewer #3: Yes: Manoel Sebastião da Costa Lima Junior

---

## [Decision Letter · Decision Letter 1]

12 Dec 2019

Dear Prof. Chatterjee,

We are pleased to inform you that your manuscript, "Iron trafficking in patients with Indian Post kala-azar dermal leishmaniasis", has been editorially accepted for publication at PLOS Neglected Tropical Diseases.

Before your manuscript can be formally accepted and sent to production you will need to complete our formatting changes, which you will receive in a follow up email. Please note: your manuscript will not be scheduled for publication until you have made the required changes.

IMPORTANT NOTES

* Copyediting and Author Proofs: To ensure prompt publication, your manuscript will NOT be subject to detailed copyediting and you will NOT receive a typeset proof for review. The corresponding author will have one final opportunity to correct any errors when sent the requests mentioned above. Please review this version of your manuscript for any errors.

* If you or your institution will be preparing press materials for this manuscript, please inform our press team in advance at plosntds@plos.org. If you need to know your paper's publication date for media purposes, you must coordinate with our press team, and your manuscript will remain under a strict press embargo until the publication date and time. PLOS NTDs may choose to issue a press release for your article. If there is anything that the journal should know, please get in touch.

*Now that your manuscript has been provisionally accepted, please log into EM and update your profile. Go to http://www.editorialmanager.com/pntd, log in, and click on the "Update My Information" link at the top of the page. Please update your user information to ensure an efficient production and billing process.

*Note to LaTeX users only - Our staff will ask you to upload a TEX file in addition to the PDF before the paper can be sent to typesetting, so please carefully review our Latex Guidelines [http://www.plosntds.org/static/latexGuidelines.action] in the meantime.

Best regards,

Syamal Roy

Guest Editor

Brian Weiss

Deputy Editor

Reviewer's Responses to Questions

Key Review Criteria Required for Acceptance?

Methods

-Are the objectives of the study clearly articulated with a clear testable hypothesis stated?

-Is the study design appropriate to address the stated objectives?

-Is the population clearly described and appropriate for the hypothesis being tested?

-Is the sample size sufficient to ensure adequate power to address the hypothesis being tested?

-Were correct statistical analysis used to support conclusions?

-Are there concerns about ethical or regulatory requirements being met?

Reviewer #1: (No Response)

Reviewer #2: Accepted. The statistical analysis specification on the threshold to define correlation between parasite load and expressed biomarkers were incorporated as it was requested.

Reviewer #3: Are the objectives of the study clearly articulated with a clear testable hypothesis? Yes

Is the study design appropriate to meet the stated objectives? Yes

Is the population clearly described and appropriate for the hypothesis being tested? Yes

Is the sample size sufficient to ensure adequate energy to deal with the hypothesis being tested? No, statistically there should be 30 people analyzed

- Was the correct statistical analysis used to support the conclusions? Correct

Are there concerns about ethical or regulatory requirements being met? No problems

Results

-Does the analysis presented match the analysis plan?

-Are the results clearly and completely presented?

-Are the figures (Tables, Images) of sufficient quality for clarity?

Reviewer #1: (No Response)

Reviewer #2: Accepted. Changes were made although the final quality of figures is an issue of the journal editors.

Reviewer #3: Does the analysis presented correspond to the analysis plan? Yes

Are the results presented clearly and completely? Yes

Are the figures (tables, images) of sufficient quality for clarity? Yes

Conclusions

-Are the conclusions supported by the data presented?

-Are the limitations of analysis clearly described?

-Do the authors discuss how these data can be helpful to advance our understanding of the topic under study?

-Is public health relevance addressed?

Reviewer #1: (No Response)

Reviewer #2: Accepted. Different questions about aspects mentioned in the original work were addressed the reading is straightforward.

Reviewer #3: Are the conclusions supported by the data presented? Yes

Are the limitations of the analysis clearly described? Not

Do the authors discuss how this data can be helpful in improving our understanding of the topic under study? Yes

Is the relevance of public health addressed? Yes, indirectly.

Editorial and Data Presentation Modifications?

Reviewer #1: (No Response)

Reviewer #2: Accept.

Reviewer #3: Change the order of description in the material and methods, first describe the population and then enter the information of the enter the information about the antibodies.

To extrapolate the conclusion of the results must be completed for 30 people for the statistical analysis to be more accurate.

Parasitic load should be expressed in parasites / µL of blood.

Summary and General Comments

Reviewer #1: (No Response)

Reviewer #2: The MS is a significant and original contribution to the relationship between iron and parasite adaptation mechanism.

Reviewer #3: The article has great potential, but needs less revision, landing is known about iron metabolism, and this work will be of great contribution.

Minor corrections should improve it, especially in the discussion.

Explain what represents the low expression CD91

PLOS authors have the option to publish the peer review history of their article (what does this mean?). If published, this will include your full peer review and any attached files.

Do you want your identity to be public for this peer review?

 For information about this choice, including consent withdrawal, please see our Privacy Policy.

Reviewer #1: No

Reviewer #2: No

Reviewer #3: No

---

## [Editor Report · Acceptance letter]

21 Jan 2020

Dear Prof. Chatterjee,

We are delighted to inform you that your manuscript, "Iron trafficking in patients with Indian Post kala-azar dermal leishmaniasis," has been formally accepted for publication in PLOS Neglected Tropical Diseases.

Best regards,

Serap Aksoy

Editor-in-Chief

Shaden Kamhawi

Editor-in-Chief
